# The Fennoscandian Shield deep terrestrial virosphere suggests slow motion 'boom and burst' cycles

Karin Holmfeldt [1✉], Emelie Nilsson [1], Domenico Simone[1,4,5], Margarita Lopez-Fernandez[1,6], Xiaofen Wu[1,7], Ino de Bruijn [2,8], Daniel Lundin [1], Anders F. Andersson [2], Stefan Bertilsson [3,9] & Mark Dopson [1]

The deep biosphere contains members from all three domains of life along with viruses. Here we investigate the deep terrestrial virosphere by sequencing community nucleic acids from three groundwaters of contrasting chemistries, origins, and ages. These viromes constitute a highly unique community compared to other environmental viromes and sequenced viral isolates. Viral host prediction suggests that many of the viruses are associated with Firmicutes and Patescibacteria, a superphylum lacking previously described active viruses. RNA transcript-based activity implies viral predation in the shallower marine water-fed groundwater, while the deeper and more oligotrophic waters appear to be in 'metabolic standby'. Viral encoded antibiotic production and resistance systems suggest competition and antagonistic interactions. The data demonstrate a viral community with a wide range of predicted hosts that mediates nutrient recycling to support a higher microbial turnover than previously anticipated. This suggests the presence of 'kill-the-winner' oscillations creating slow motion 'boom and burst' cycles.

[1] Centre for Ecology and Evolution in Microbial Model Systems (EEMiS), Linnaeus University, Kalmar, Sweden. [2] Department of Gene Technology, Science for Life Laboratory, KTH Royal Institute of Technology, Stockholm, Sweden. [3] Department of Ecology and Genetics, Limnology and Science for Life Laboratory, Uppsala University, Uppsala, Sweden. [4] Present address: Department of Plant Biology, Swedish University of Agricultural Science, Uppsala, Sweden. [5] Present address: SLU Bioinformatics Infrastructure, Swedish University of Agricultural Science, Uppsala, Sweden. [6] Present address: Helmholtz-Zentrum Dresden-Rossendorf e.V., Institute of Resource Ecology, Dresden, Germany. [7] Present address: Center for Environmental Biotechnology, University of Tennessee, Knoxville, TN, USA. [8] Present address: Memorial Sloan Kettering Cancer Center, New York, NY, USA. [9] Present address: Department of Aquatic Sciences and Assessment, Swedish University of Agricultural Sciences, Uppsala, Sweden. ✉email: karin.holmfeldt@lnu.se

The 'deep biosphere' is the vast amount of life that exists under the Earth's surface including, water-filled bedrock fractures[1]. Despite being predominantly highly oligotrophic, the deep biosphere represents the largest organic carbon pool on Earth[2,3] and is estimated to contain up to $6 \times 10^{29}$ microbial cells[4]. Deep terrestrial biosphere communities host members of all three domains of life that are inferred to be active based on, e.g., RNA transcript profiles[5–7] and video recordings of eukaryotic communities in fissure systems[8]. Despite the suggested size and importance of this biome, largely due to the difficulty in obtaining uncontaminated samples[9], it is one of the least understood environments on Earth.

Viruses represent the most diverse and abundant biological entities known on Earth, but even the most common environmental viruses lack isolated representatives[10,11]. Deep terrestrial biosphere viral abundances range from $10^4$ to $10^6$ particles mL$^{-1}$ with a maximum virus:bacteria ratio of 7:1, as previously reported[12]. Further support for a deep terrestrial virosphere comes from observations of viral sequences in a microbial metagenome from a Finnish 2516-m-deep drill hole[13]. In addition, a 2.5-km-deep shale ecosystem contains a viral community in which CRISPR arrays enabled links to be made between viral contigs and microbial genomes[14]. Deep biosphere viruses influence microbial community dynamics by releasing nutrients that would otherwise be bound in living cells[15,16] and have been shown to mediate horizontal gene transfer in *Candidatus* Desulforudis audaxviator cells captured from 3 km deep groundwater[17]. However, no studies have reported RNA transcripts for viral particle propagation in deep terrestrial biosphere host cells.

An additional terrestrial site is the 460-m-deep, 3.6-km-long tunnel making up the Swedish Nuclear Fuel and Waste Management Company (SKB) operated Äspö Hard Rock Laboratory (Äspö HRL). The Äspö HRL is located in 1800 million-year-old Fennoscandian Shield bedrock containing fractures bearing groundwaters of varying characteristics dependent on age and connectivity to the surface[18,19]. Bacteria found in the Äspö HRL have been extensively characterized by culture-dependent and 16S rRNA gene sequence methods[20–24] while microbial metagenomes from the same waters analyzed in this study uncovered a diverse community fed by organic carbon or hydrogen[25]. Furthermore, RNA transcripts from Äspö HRL groundwaters with higher relative concentrations of organic carbon showed an active community with a range of metabolic strategies. In contrast, the low organic carbon water has the potential to carry out translation but only replicates when a carbon and energy source is intermittently available, a state defined as 'metabolic standby'[6,7]. Kyle et al.[26] reported that the number of virus-like particles in the Äspö HRL groundwaters range from $10^5$ to $10^7$ particles mL$^{-1}$ and include a lytic *Podoviridae* virus infecting the sulfate-reducing bacterium *Pseudodesulfovibrio aespoeensis* (formerly *Desulfovibrio aespoeensis*)[27]. To date, no broader characterization of the viral communities and (meta)genome structure has been reported for the Äspö HRL groundwaters.

Due to the lack of knowledge regarding the diversity and activity of viruses within the deep biosphere, we aim to investigate this through viromes from pristine continental deep biosphere groundwaters. This includes two modern marine groundwaters from the Äspö HRL at different depths (171.3 and 415.2 m below sea level (mbsl)) and one thoroughly mixed groundwater from the Äspö HRL (448.4 mbsl). Our hypotheses were that (1) due to similar geochemistry and age[25], the two Äspö HRL modern marine groundwater viromes are more similar to each other than to the thoroughly mixed water and (2) that similarly to the bacterial community[6,7], the more shallow virome is more active than the deeper groundwaters. The data provide insights into the viral populations, potential hosts, and relationship to other viral communities for three water types of varying characteristics. Finally, community RNA transcripts demonstrate viral activity within the deep terrestrial biosphere.

## Results and discussion

The 3.6-km-long Äspö HRL tunnel originates on the Baltic Sea coast in southeastern Sweden and extends under both seawater and coastal land. Accordingly, meteoric water, modern Baltic Sea water, and ancient old saline waters influence the groundwaters at varying levels below the surface. The three boreholes sampled in this study extend from the tunnel and intersect fissure systems containing anoxic groundwaters of different ages and origins[24,25]. The first two studied boreholes were SA1229A-1 at 171.3 mbsl and KA3105A-4 at 415.2 mbsl that both contain infiltrated Baltic Sea brackish water of <20 years and were termed 'modern marine' (designated 'MM-171.3' and 'MM-415.2', respectively). The final borehole was KA3385A-1R at 448.4 mbsl that is a 'thoroughly mixed' groundwater containing old saline deep groundwater from a few thousand years ago that is diluted with waters such as glacial meltwater from the most recent Pleistocene continental ice sheet along with water infiltrating from the Littorina Sea a few thousands of years ago (termed 'TM-448.4'). The MM-171.3, MM-415.2, and TM-448.4 groundwaters have previously been defined as MM, UM, and OS in Wu et al.[25], respectively.

**Identification of viral contigs from metagenomic data.** The Äspö HRL groundwaters passing through the 0.22-μm filter used for cell capture contained both small-celled bacteria and viruses[25]. Therefore, the corresponding metagenome dataset could not be defined as exclusively 'viral'. In addition, this filtration step will have removed viruses adsorbed to cells, intracellular and integrated viruses, and viruses larger than 0.22 μm. Bioinformatics tools were thus used to predict if contigs containing >10 genes were of putative viral origin. By comparing different cut-off values for 'cellular' versus 'non-cellular' sequences and the occurrence of a specific contig in either a viral or a microbial bin, a cut-off of <70% cellular genes was chosen to define a contig to be of likely viral origin (Supplementary Fig. 1). Using this cut-off on the six metagenomes isolated from Äspö HRL groundwaters yielded a total of 3957 putative viral contigs, with the fewest predicted in the MM-171.3 and the most in the MM-415.2 (Supplementary Table 1). This collection was expanded with 94 additional contigs detected by VirSorter (see Supplementary Data 1 for details of each contig). As the metagenomes consisted of both cellular organisms and viruses in combination with the exclusive use of contigs encoding >10 genes, a conservative estimate of 1.7–6.5% of the total metagenomes were confirmed as 'viral' and used for further analyses (Supplementary Table 1). These values were especially low for the MM-171.3 groundwater, where only 1.7–1.9% of the reads could be mapped to viral contigs compared to 41–46% of the reads mapping to microbial metagenome-assembled genomes (MAGs; with 9–16% of the dataset in MAGs for the other groundwaters). These percentages were unaffected by the sequencing coverage assessed using Nonpareil as the metagenomes showed metagenomic coverage of 66–86% (Supplementary Table 1). The higher percentage of reads assigned to microbial MAGs in the MM-171.3 compared to the other Äspö HRL groundwaters supports instead that a larger proportion of cells passed the 0.22-μm filter used during cell capture and consequently, this groundwater had a higher proportion of ultra-small bacterial cells[25,28].

Between 40 and 44% of the putative viral ORFs showed similarity (e-value cut-off: 0.001) to proteins in NCBI nr protein database (Supplementary Table 1). This is a higher number than is typically seen among viral metagenomes[29,30] and more similar

to numbers seen for sequence data of aquatic phage isolates[31,32]. It should be noted that the Äspö HRL analysis is based on complete contigs, resembling viral genomes, instead of individual reads and thus only includes a small subset of the complete data. Even though the metagenomic contigs had gone through rigorous controls to define them as viral, the majority of the best alignments were taxonomically affiliated with cellular organisms (Supplementary Table 1). Such high proportions of ORFs with homologs in cellular organisms are common both in viral genomes[31] and viral metagenomes[29]. This is largely due to the lack of closely related environmental viral sequences in databases and consequently viral ORFs often have their best alignment to viral proteins integrated into bacterial genomes by temperate

viruses, proteins moved through horizontal gene transfer, and viral encoded host genes[33].

**Unique deep subsurface viral communities.** Network analysis based upon amino acid similarities[34] among the predicted proteins within each contig in the Äspö HRL viral metagenomes and the genomes of prokaryote viral isolates in NCBI RefSeq showed very few connections (Fig. 1 and Figshare[35]). While viral metagenomes often show low similarity to sequenced isolates, comparison of different viral metagenomes commonly shows larger resemblance[29]. However, no Äspö HRL proteins clustered with the Tara Oceans global viral metagenome protein clusters[36]. Using a less stringent method, <30% of the Äspö HRL viral

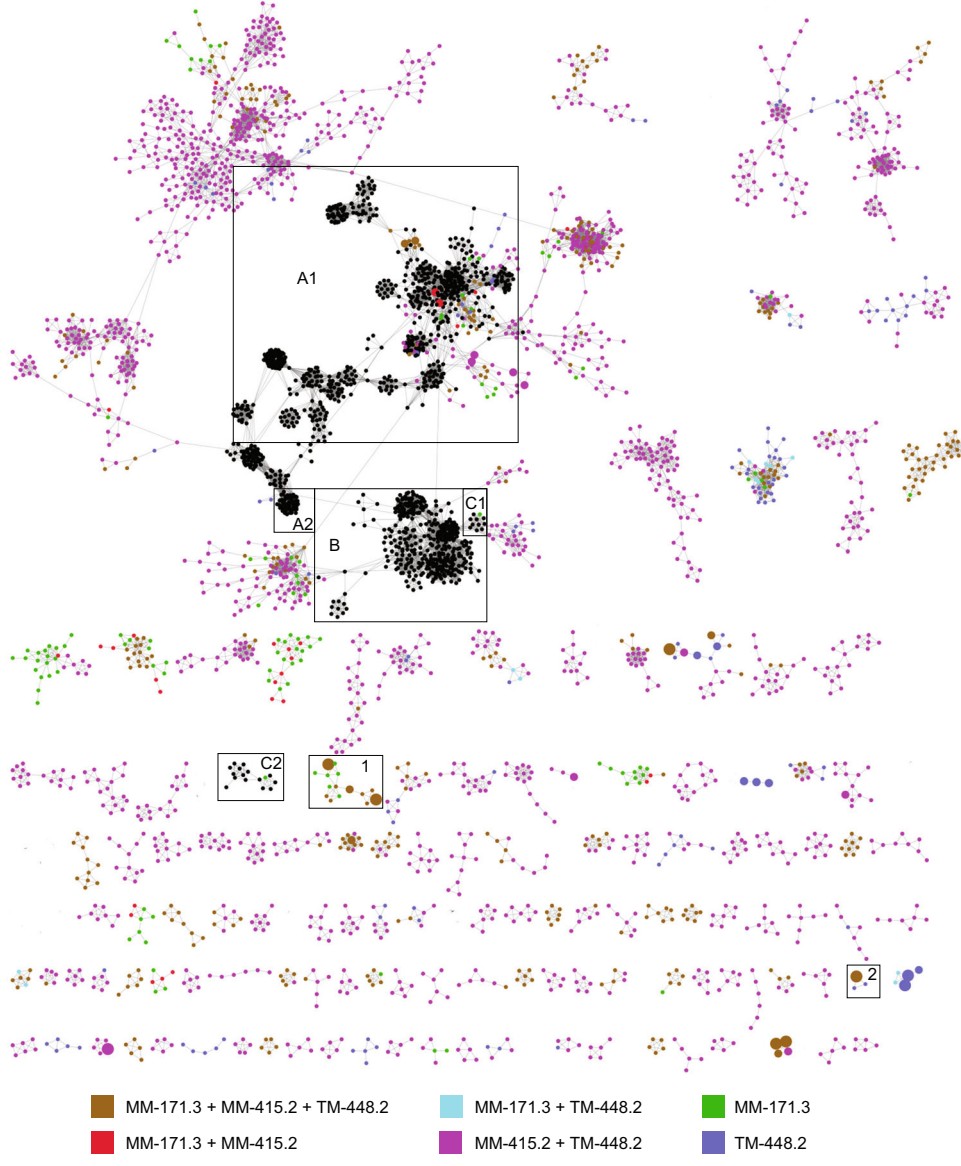

| MM-171.3 + MM-415.2 + TM-448.2 | MM-171.3 + TM-448.2 | MM-171.3 |
| MM-171.3 + MM-415.2 | MM-415.2 + TM-448.2 | TM-448.2 |

**Fig. 1 Network displaying Äspö HRL contigs and viral isolates highlighting connections between groundwaters.** A selected portion of the network analysis of the contigs from the three groundwaters along with sequences from bacterial and archaeal viral isolates in the NCBI RefSeq database (black nodes). Color coding of nodes represents the presence of the contigs in the groundwaters based on metagenome read recruitment. Boxes marked with letters refer to phages that mainly infect Gamma-, Beta-, and Alphaproteobacteria (A1 & A2); Firmicutes (B); and Bacteriodetes (C1 & C2). Boxes marked with numbers refer to clusters consisting of viral contigs both in the core and in individual groundwaters. Node sizes for Äspö HRL viral contigs are based on bp reads recruited divided by the contig size in kb and the metagenome size in Mb (smallest circle <50, medium circle 50–100, and largest circle >100 bp mapped/kb of contig/Mb of metagenome) for the metagenome from which the contig originated (see Supplementary Fig. 3 for information regarding the groundwater type where a specific contig originated). Reference viral isolates in the NCBI RefSeq database are all presented using the smallest circle size and do not represent abundance. The complete and unedited Cytoscape figure is available in Figshare[35].

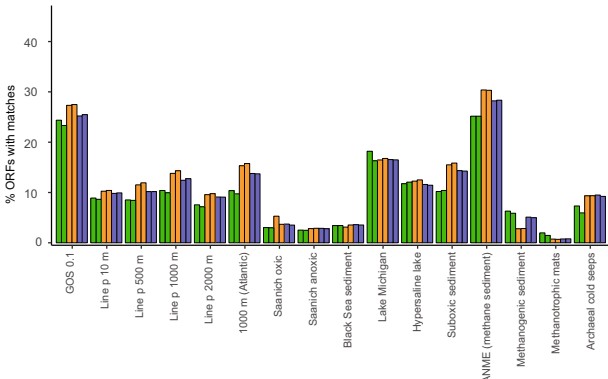

**Fig. 2 Similarity of Äspö HRL ORFs to metagenomes from other environments.** Percent of ORFs in the duplicate MM-171.3 (green), MM-415.2 (orange), and TM-448.2 (blue) groundwater metagenomes that share protein similarities (BLASTp; e-value cut-off 0.001) to viral sequences in selected environmental viral metagenome datasets from the published literature. Details and accession numbers for each metagenome are in Supplementary Data 2.

proteins could be aligned against 16 viral metagenomic datasets collected from various environments (Fig. 2 and Supplementary Data 2). In addition, comparison of the Äspö HRL proteins to three deep biosphere metagenomic assembled viral genomes only gives spurious matches[37] (Supplementary Data 1). Finally, mapping of metagenomic reads from 25 Baltic Sea viral metagenomes to the Äspö HRL viral contigs showed that only 49 out of 4051 viral contigs (~1%) were present in any Baltic Sea metagenome (Supplementary Data 3, sheet recruit. LMO viral metag.). Despite a high proportion of cellular organisms within the Äspö HRL viral metagenome, recruitment of reads from seven Baltic Sea microbial metagenomes to all Äspö HRL contigs showed that <0.2% of contigs were covered with reads covering >75% of the contig length (Supplementary Data 3, sheet recruit. LMO cell. metag.). Based on the lack of similarity between the Äspö HRL viral sequences and all investigated datasets, we suggest that the three investigated Äspö HRL groundwaters feature largely unique viral communities. This finding was remarkable since MM-171.3 and MM-415.2 groundwaters have a residence time of <20 years and are fed by infiltration of Baltic Sea water likely also transporting microorganisms into this deep groundwater[24]. This suggests a cell sorting process has occurred between marine surface water microorganisms, through shallow sediments that consist of a selection of microbes found in surface sediments[38], to deep biosphere communities. This also provides support for the view that this cell sorting in terrestrial deep biosphere communities selects for populations adapted to the highly oligotrophic conditions of this biome[7,39].

**Viral distribution within the three groundwaters.** Reads from all surveyed metagenomes were individually recruited to the viral contigs (see Supplementary Data 1 for details of each contig). To be considered as present in a sample, at least 75% of a contig was required to be covered by metagenomic reads[36]. Based on this, 15% of the contigs were shared among all groundwaters while 70% of the contigs were shared between the MM-415.2 and TM-448.2 groundwaters (Supplementary Fig. 2). This high similarity among the viral communities in the MM-415.2 and the TM-448.2 groundwaters is also evident in the contig-based network analyses (Fig. 1). On the other hand, both the MM-171.3 and TM-448.2 viral communities also had unique members, represented by 235 and 304 contigs, respectively, while only one viral type (represented by two contigs) was truly unique for MM-415.2 (Supplementary Fig. 2).

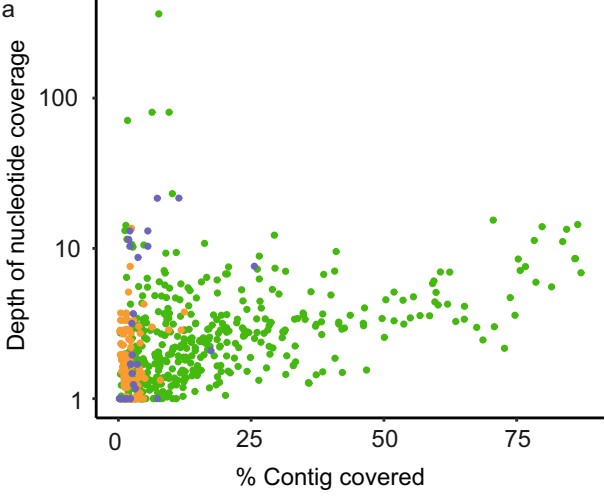

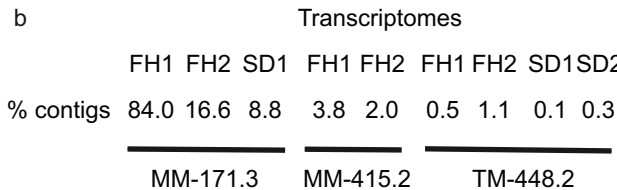

**Fig. 3 Transcription-based activity of Äspö HRL viral contigs.** Summary of read mappings for the MM-171.3 (green), MM-415.2 (orange), and TM-448.2 (blue) groundwater metatranscriptomes to viral contigs (**a**). The values in (**b**) give the percentage of contigs recruiting reads (at least one read per contig) from the replicate metatranscriptomes ($n = 2$ to 4) for the three groundwater types.

The core viral community (contigs occurring in all three groundwaters) were highly diverse and were included in network-clusters ranging from only core-clusters to clusters composed of viruses both in the core and those occurring in only one groundwater (Fig. 1, Supplementary Fig. 3, and Figshare[35]). Several of the most abundant viruses in the Äspö HRL were present within the core, but often they were only highly abundant in the groundwater they were originally assembled from and occurred in low abundances in the other groundwaters. For instance, the core MM171B00549, MM171A01743, and MM171B01602 viruses were highly abundant in MM-171.3 (Fig. 1, cluster 1) and TM448A02179 was highly abundant in TM-448.2 (Fig. 1, cluster 2).

Overall, the data points towards a partially shared viral community between the different water types and a far-reaching exchange of viruses between the MM-415.2 and TM-448.2 communities. This is supported by mixing of groundwaters within the Äspö HRL and the largely overlapping microbial communities seen in these two groundwater types[25].

**Transcript-based viral activity.** Community RNA was extracted from deep biosphere cells captured utilizing filter holders[6] and a specially designed sampling device[7] for the MM-171.3 and TM-448.2 groundwaters and only with filter holders for the MM-415.2 groundwater. Metatranscriptomic reads from each groundwater were recruited to viral assigned contigs from the metagenomes. Up to 84% of the viral contigs were transcribed with up to 87% coverage in the MM-171.3 groundwater compared to transcripts only mapping to <4% and 1% of the viral contigs in the MM-415.2 and TM-448.2 groundwaters, respectively (Fig. 3, Supplementary Data 1 for details of each contig). These metatranscriptomes represent 'snapshots' sampled at different time

points than when the viral metagenomes were generated and they are only compared to a subset of the entire viral community (represented by the metagenomically assembled contigs) lacking, e.g., prophages removed in the metagenomes by cell capture on the 0.2 μm filters. Therefore, we cannot rule out that additional, unidentified viruses in the MM-415.2 and TM-448.2 groundwaters were active at the time of cell capture. However, as the MM-171.3 groundwater has an active, replicating microbial community while the TM-448.2 community appears to be in "metabolic standby"[7], the most plausible explanation is that the viral community in the MM-171.3 is also actively replicating while viruses in the MM-415.2 and TM-448.2 groundwaters largely lack the required replicating hosts in order to be active.

**Viral taxonomic affiliation and potential host.** The similarity of the Äspö HRL viral metagenome to cellular organisms in the NCBI nr database can be used to infer the potential hosts for viruses that are solely represented by metagenome sequences[40]. Among the viral proteins that aligned to cellular organisms, the majority were to Bacteria (34–39% of all ORFs). Of these, viral protein matches to Gammaproteobacteria, Patescibacteria, Alphaproteobacteria, Firmicutes, and Chloroflexota dominated in all groundwaters (>2% per phylum; Fig. 4a). On a community level, these results agree with the contig-based host predictions using k-mer analyses, for which 925 contigs could be assigned to a potential host on phylum level (Fig. 5, Figshare[35], and Supplementary Data 1). Of the viral contigs for which a host was predicted at the phylum level, 62% (574 contigs) were suggested to be affiliated to Firmicutes (contigs over 40 kb can be seen in Supplementary Fig. 4). However, the network analysis showed few connections of Äspö HRL viral contigs to known, isolated Firmicutes phages (Figs. 1 and 5). While Firmicutes are known to be abundant in other deep subsurface environments[17], bacteria of this phylum only contributed to a very minor relative proportion of the MM-171.3 and TM-448.2 communities as investigated by metagenomics[25] and 16S rRNA gene amplicon sequencing[24].

Conversely, bacteria within Firmicutes[6,7] and their viruses (Supplementary Data 1) appear to be disproportionately active in Äspö HRL fracture waters.

To provide insights into the taxonomic affiliation of the Fennoscandian Shield deep virosphere to isolated sequenced viruses, the metagenome-coded proteins were aligned (BLASTp) to viral genomes in the NCBI RefSeq database. The results gave significant alignments for 22–25% of the proteins, with the majority of best alignments to members of *Caudovirales* including (from greatest to least number of hits) *Siphoviridae*, *Myoviridae*, and *Podoviridae* (Fig. 4b). However, the dominance of hits to *Caudovirales* should be interpreted with caution due to the overrepresentation of such viral genomes in the NCBI database. In the majority of cases, the similarity to known viruses appears to be only spurious alignments of single genes providing little or no information of the relatedness of the viruses within the Äspö HRL groundwaters to known, isolated viruses (Fig. 1). For those few Äspö HRL viral contigs that showed similarity to isolated viruses, the majority were to phages that infect various Proteobacterial hosts (Fig. 1 and Supplementary Fig. 5). Furthermore, Gammaproteobacteria were also a commonly predicted host for the Äspö HRL viral contigs (54 contigs; Fig. 5) and this bacterial class is reported to be both present and active within these fracture waters[7,25]. The only cultured Äspö HRL phages are five unsequenced podoviruses that infect the Desulfobacteriota species *Pseudodesulfovibrio aespoeensis*[27]. Desulfobacteriota was the fifth most common predicted host (predicted host for 38 contigs) for the viruses within the investigated Äspö HRL groundwaters and these viral contigs were not associated with previous known viral isolates (Fig. 5). They were mainly detected in the two deeper groundwaters but showed neither particularly high abundance nor activity (Supplementary Fig. 6).

Forty-nine diverse viral clusters lacking associations to known viral isolates were suggested to infect hosts from the Candidate Phyla Radiation[41] within the Patescibacteria superphylum[42] (Fig. 5). Reflecting the large host diversity, viruses infecting the same

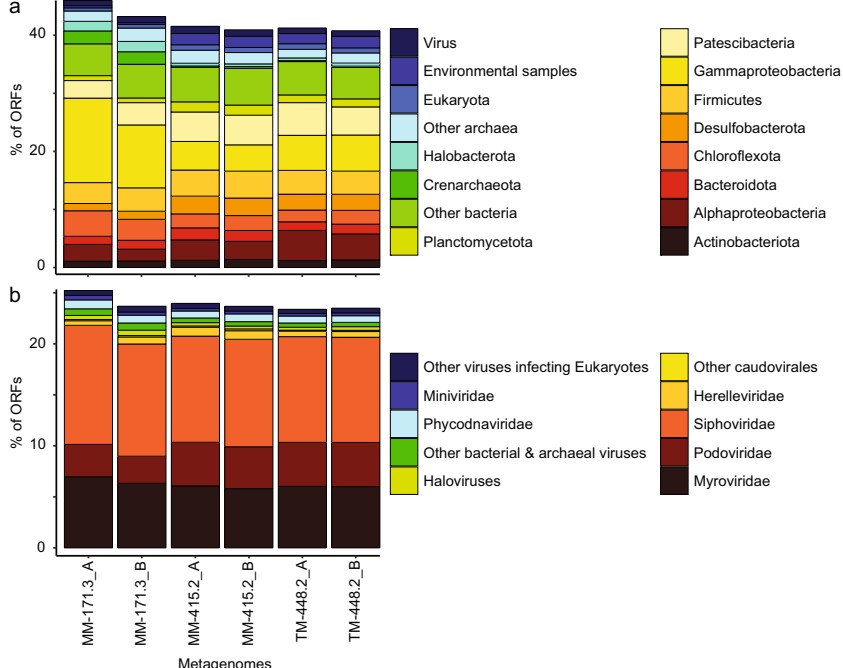

**Fig. 4 Similarity of Äspö HRL viral ORFs to isolated viruses and cellular organisms.** Percentage of Äspö HRL viral ORFs matching taxonomic units in **a** NCBI nr (June 2017) and **b** ViralDB (July 2019). Taxonomic units comprising <1% of the matches to NCBI nr and <0.2% of the matches to ViralDB are placed in 'others'.

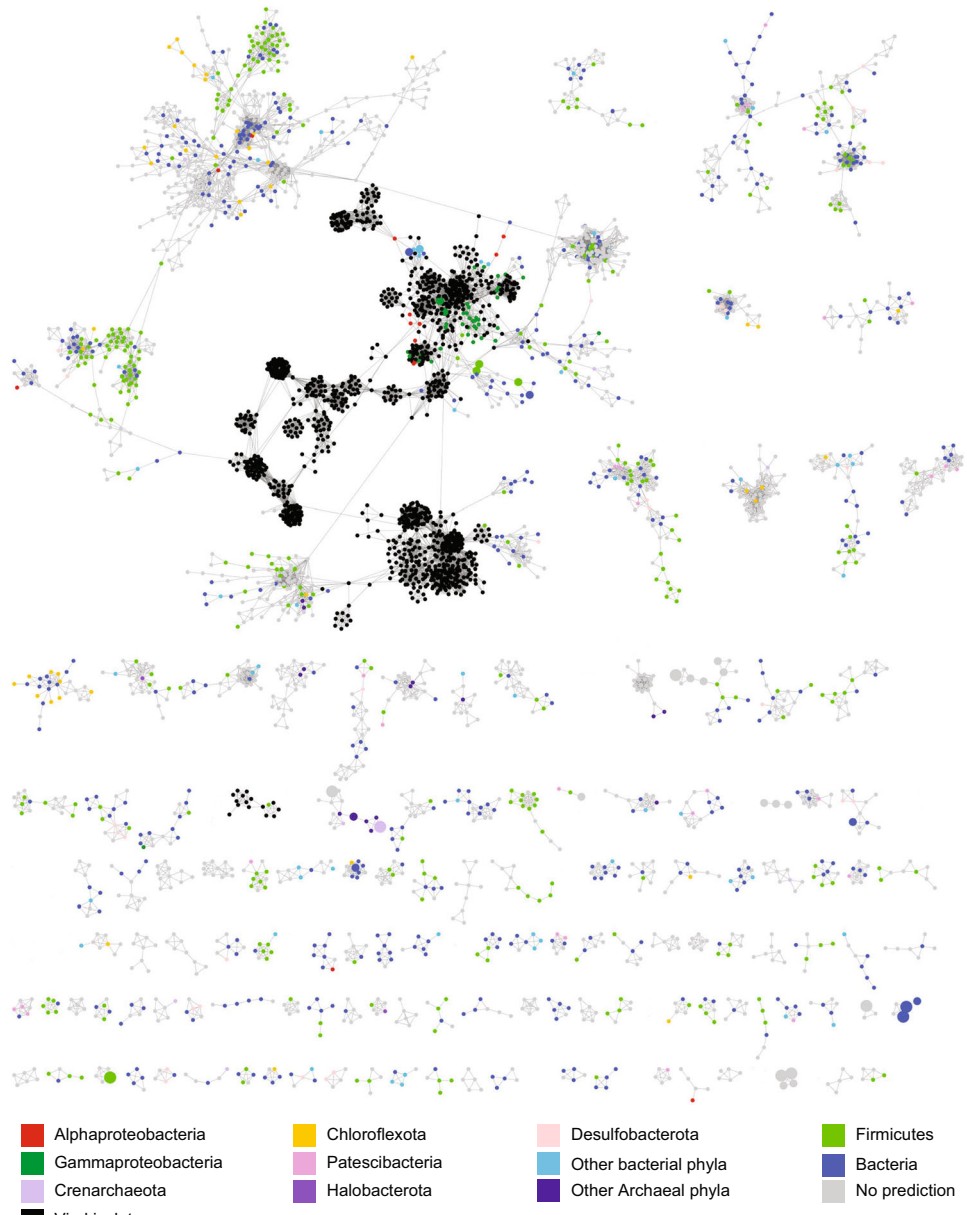

| | | | |
|---|---|---|---|
| 🟥 Alphaproteobacteria | 🟨 Chloroflexota | 🟪 Desulfobacterota | 🟩 Firmicutes |
| 🟩 Gammaproteobacteria | 🟪 Patescibacteria | 🔵 Other bacterial phyla | 🔵 Bacteria |
| 🟪 Crenarchaeota | 🟪 Halobacterota | 🟪 Other Archaeal phyla | ⬜ No prediction |
| ⬛ Viral isolates | | | |

**Fig. 5 Network displaying Äspö HRL contigs and isolated viruses highlighting potential hosts.** A selected portion of the network analysis of the contigs from the MM-171.3, MM-415.2, and TM-448.2 groundwaters along with sequences from bacterial and archaeal viral isolates in the NCBI RefSeq database. Colored nodes represent predicted microbial hosts based upon k-mer analysis of the Äspö HRL viral contigs (full data in Supplementary Data 1). Node sizes are as described in Fig. 1. The complete and unedited Cytoscape figure is available in Figshare[35].

Patescibacteria phyla also show large genetic differences (Fig. 6). To the best of our knowledge, this is the first report of active viruses infecting these, so far uncultured, bacteria that are prevalent and play an important role in biogeochemical cycling in the terrestrial deep biosphere[23,24,43,44]. For example, these taxa include Yanofs-kybacterales and Gottesmanbacteria that have reduced genome sizes as an adaptation to the highly oligotrophic conditions along with Chisholmbacteraceae and Zambryskibacteraceae that are involved in cryptic cycling of nitrogen and/or sulfur[41,42,45]. Many of the potential Patescibacteria viruses were lacking in the MM-171.3 groundwater while all except one were present in either (and often both) of the MM-415.2 or TM-448.2 groundwaters (Fig. 6 and Supplementary Data 1). This suggests a larger diversity of Patescibacteria viruses in the deeper groundwaters.

The MM-171.3 groundwater viral contigs had 6% best protein alignments to Archaea compared to 2–3% in the MM-415.2 and TM-448.2 groundwaters of which the dominant matches were to Crenarchaeota and Halobacterota (Fig. 4). In addition, viruses (represented by contigs) predicted to infect Archaeal hosts were particularly abundant (Fig. 5 and Supplementary Fig. 7) and seemingly active (Supplementary Data 1) in the MM-171.3 groundwater. In addition, several of the Archaea viruses appeared to be uniquely occurring either in only MM-171.3 groundwater or in both MM-415.2 and TM-448.2. Although Archaea are not known to be abundant or active within the Äspö HRL fracture waters[6], archaeal viral activity might point to a more important role of this domain within this deep subsurface environment than previously thought.

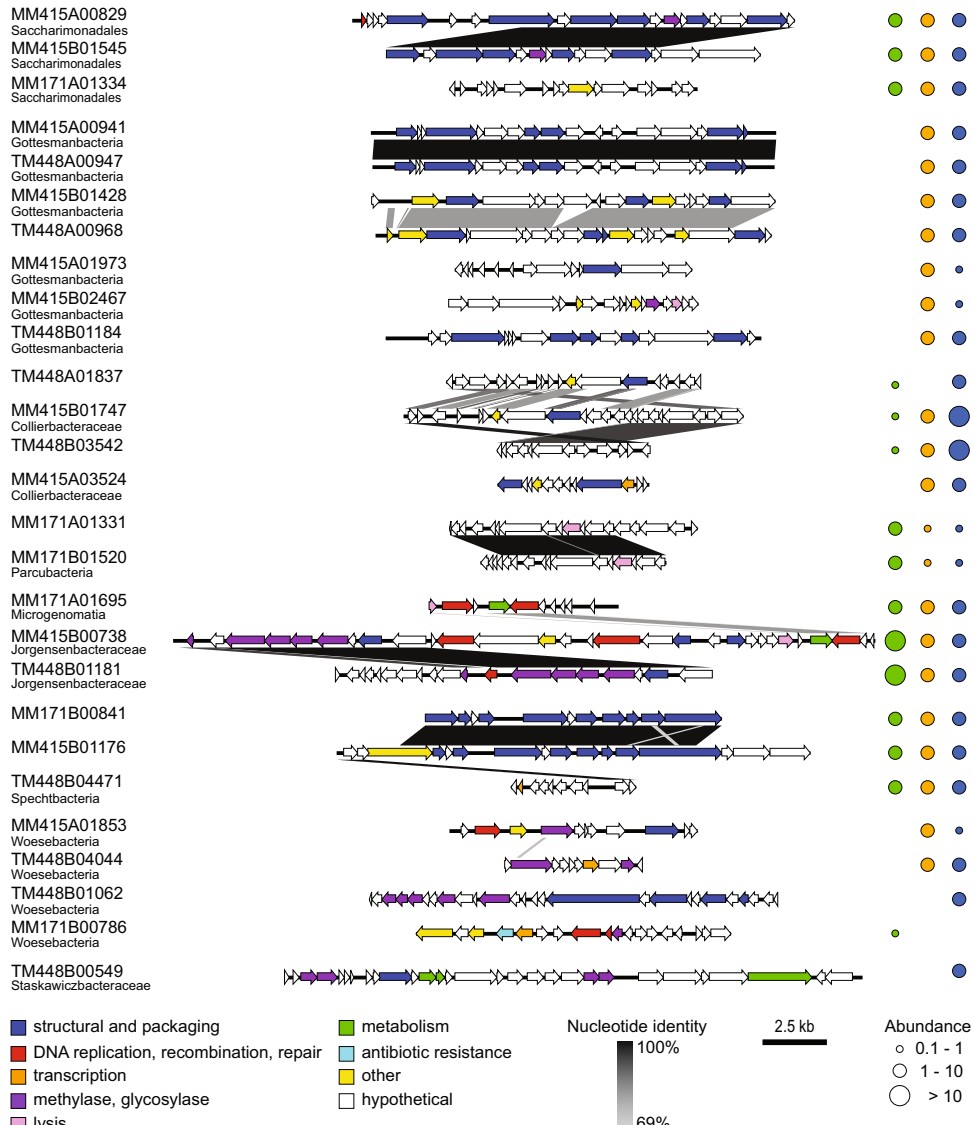

**Fig. 6 Annotation of viral contigs potentially infecting Patescibacteria and their relative abundance within the Äspö HRL groundwaters.** Selected viral contigs suggested by k-mer analysis to infect Patescibacteria where lower taxonomic ranks could be determined as well as contigs similar to those in the three Äspö HRL groundwaters. Functional annotation of the predicted genes is given by color coding, nucleotide identity between contigs is given by the heat scale, the scale bar represents contig length, and abundance is given as mean base pair read depth normalized for the metagenome size in the MM-171.3 (green), MM-415.2 (orange), and TM-448.2 (blue) groundwaters.

The large number of disparate contig clusters not connected to sequenced viral isolates and the lack of predicted hosts likely reflects the poorly investigated deep terrestrial biosphere microbial community. In some cases, the relative abundance and number of predicted host and viral transcripts is explained by predation of viruses on their predicted hosts. For instance, the low Firmicutes abundance may be due to the large diversity, abundance, and activity of predating phages (Fig. 5 and Supplementary Data 1). This host versus phage relationship has previously been observed for some low abundance marine bacterial taxa[46]. However, this relationship is not always observed and could be because the viral metagenomes and metatranscriptomes were sampled at different time points and may simply reflect viral predation on dominant microbial populations creating 'boom and burst' cycles between the viruses and their predicted microbial hosts.

**Metabolic functioning of the Äspö HRL viromes.** Overall, the dominance of structural genes as well as genes involved in DNA

replication and modification supports the viral origin of the data (see Supplementary Data 4 for functional annotation of all predicted proteins). As in other environments[29,47], metabolic genes seem to be of importance within the Äspö HRL viral community. This includes genes involved in sulfur cycling that reflect the significance of this element in the deep biosphere. Genes involved in sulfur cycling showed a clear partitioning between the viral metagenomes with phosphoadenosine phosphosulfate (PAPS reductase) and adenylyl-sulfate kinase (APS kinase) involved in sulfate assimilation predominantly identified in the MM-415.2 and TM-448.4 groundwaters. In contrast, sulfatase (soxB) that is part of the thiosulfate oxidizing Sox complex was predominantly identified in actively transcribed viral contigs in MM-171.3. This partition likely reflects the highly oligotrophic conditions with little organic material in the TM-448.4 groundwater compared to the more active community suggested to couple inorganic sulfur compound oxidation to, e.g., nitrate reduction assigned to Gammaproteobacteria and Cyanobacteria

in MM-171.3[25]. A further metabolic gene identified in all three metagenomes but mostly occurring in actively transcribed viruses in MM-171.3 was a putative zinc- or iron-chelating ferredoxin protein potentially involved in a wide variety of electron transfer reactions in, e.g., Proteobacterial classes, Patescibacteria, and Euryarchaeota. In addition, viruses potentially infecting Firmicutes contained carbamoyltransferase genes for cephamycin synthesis in contigs from all three investigated fracture waters. The potential transfer of genes for cephamycin synthesis would be important in the deep biosphere as this antibiotic is highly effective against anaerobic microorganisms[48]. A further nine contigs encoded the cephalosporin antibiotic biosynthesis protein cephalosporin hydroxylase. These included MM-171.3 viruses potentially infecting Gammaproteobacteria as well as contigs from MM-415.2 and TM-448.4 that represented viruses potentially infecting Firmicutes. Antibiotic resistance genes were also present in all three boreholes. These include *femAB* involved in methicillin resistance that was present in contigs in all three fracture waters but only in active viruses in MM-171.3 and *bacA* involved in bacitracin resistance in contigs with a predicted Patescibacteria host, which was only present and actively transcribed in MM-171.3 (Fig. 6 and Supplementary Data 1). In contrast, a *marA2* multiple antibiotic resistance gene was only present in contigs in the MM-415.2 and TM-448.4 boreholes.

The presence of viral coded metabolic genes within the subset of viruses actively replicating within the Äspö HRL groundwaters indicated potential transfer of functional capacities between populations, broadly increasing the fitness within the microbial community. Virus genes coding for antibiotic production and resistance also suggested that in spite or even due to the highly oligotrophic conditions, at least a portion of the viruses in the deep biosphere promote competition for the limited resources and available space for growth.

**Conclusions**. The Äspö HRL deep terrestrial biosphere contains a diverse and previously overwhelmingly unknown viral community with a wide range of predicted hosts, including taxa such as the Patescibacteria, for which we previously lacked knowledge about active phages. Despite that the two modern marine groundwaters have similar water characteristics, the MM-415.2 and TM-448.2 viromes were more similar to each other than MM-171.3 to MM-415.2 and thus, refuting hypothesis one. The reason for the higher resemblance of viromes from MM-415.2 and TM-448.2 is yet unclear but may reflect the greater depth from the surface for these two groundwaters. The presence of RNA transcripts predominantly in the MM-171.3 validated hypothesis two of an actively replicating microbial community in the shallow groundwater compared to the community in the thoroughly mixed groundwater, which is in 'metabolic standby'[7]. The occurrence of viruses suggests deep biosphere nutrient recycling that would support a higher rate of microbial replication than previously anticipated and indicates the presence of a 'boom and burst' cycle in slow motion. Finally, the turnover of the dominant microbial population in a borehole observed over an eight month-period at the Äspö HRL[49] could potentially have been due to viral predation according to the 'kill-the-winner' model.

## Methods

**Äspö HRL and water types**. The Äspö HRL tunnel is located in 1800 million-year granite and quartz monzodiorite on the southeastern coast of Sweden (Lat N 57° 26′ 4″ Lon E 16° 39′ 36″) and its geology, hydrology, and chemistry have all been previously published[50–52].

**Sampling of viral particles, DNA preparation, and sequencing**. Sampling of the three boreholes was conducted in April–May 2013 and was carried out by flushing three to five section volumes to ensure stagnant water had been expelled and subsequently collecting water samples in sterile plastic containers. Duplicate sets of containers from the three separate groundwaters were transported to the laboratory and filtered through a 0.22 μm membrane filters (47 mm: cellulose nitrate, Sartorius; 142 mm: Isopore polycarbonate, Millipore) within <2 h to give biological replicates ($n = 2$) for each water type. Viral particles in the filtrate were precipitated with $FeCl_3$ (final concentration 1 mg L$^{-1}$, Sigma) and collected on 0.8-μm filters (142 mm; Isopore, Millipore) according to the iron-chloride precipitation method[53]. The filters were stored in Falcon tubes at +4 °C until DNA extraction. For DNA extraction, the viral particles were resuspended in ascorbate solution (20 mL 0.5 M EDTA pH 8, 12.5 mL 1 M Tris pH 8, 1.9 g MgCl$_2$, 3.52 g ascorbic acid, ~4 mL NaOH, and MilliQ ultrapure water up to 100 mL) and concentrated using 50 kDa Amicon spin-filters (Millipore). The viral particles were treated with DNase I (50 U, Thermo Scientific) at 37 °C for 30 min and EDTA (0.15 M) at 65 °C for 15 min, followed by pre-heated (37 °C for 30 min) proteinase K (2 mg/mL; Fisher) at 37 °C for 12 h and the DNA was then extracted with the Wizard PCR DNA Purification kit (Promega). Here, 1 mL resin was added to 1 mL of the viral suspension, thoroughly mixed, and then pushed through the minicolumn. The column was washed twice using 1 mL 80% isopropanol and residues of isopropanol were removed by centrifugation (2 min at 10,000 × g). The DNA was eluted by adding 100 μL pre-heated (80 °C) 10 mM Tris (Trizma base (Sigma), pH 8) to the column followed by vortexing (10 s) and centrifugation (10,000 × g for 30 s). Quality and quantity of the DNA were verified with Nanodrop (Thermo Scientific) and Qubit (high sensitivity DNA kit; Invitrogen, Life Technologies), respectively.

The Illumina sequencing libraries were prepared with the Takara ThruPlex DNA-seq Kit and duplicate metagenomes from the three water types were sequenced on two Illumina HiSeq flow cells (HCS2.0.12.0/RTA 1.17.21.3; i.e., one metagenome from each borehole in each cell) with a 2 × 101 setup in High Output mode. However, it should be noted that the ThruPlex DNA-seq Kit includes an amplification step that can potentially add bias to the sequencing libraries. Details of the volume of water sampled as well as the quantity and quality of the extracted DNA are given as the <0.22 μm dataset in Wu et al.[25] while sequencing and quality control data are in Supplementary Table 1. Raw reads are deposited in NCBI, BioProject ID PRJNA279924.

**Identification of viral contigs**. Quality control of the sequenced reads was conducted using FastQC (version 0.11.2) before removing the adapters and low-quality reads using Trimmomatic[54]. The trimmed reads were assembled using the de novo assembler Ray (version 2.3.1 and version 2.3.0)[55] with k-mer sizes of 31, 41, 51, 61, 71, and 81. Newbler (version 2.6) was then used to merge all the contigs generated from different k-mers. Nonpareil was run on all metagenomes individually (version 3.304, settings: -T k-mer -f fastq) to estimate the sequencing coverage of the community[56]. All contigs >700 bp were uploaded to Metavir (http://metavir-meb.univ-bpclermont.fr) that provided the ORF-calling. Three methods were used to define contigs of putative viral origin (i.e., non-cellular). Firstly, all contigs were placed into bins based on k-mer frequencies and relative abundance with cellular contigs being defined on the presence of microbial single copy genes[25]. Secondly, contigs containing >10 ORFs were divided into 'cellular' or 'non-cellular' depending on the percentage of best alignments for the ORFs (Diamond-BLASTp, v 0.8.26; e-value cut-off 0.001, against NCBI non-redundant database; June 2015) to Bacteria, Archaea, or Eukarya reference sequences (defined as 'cellular') or to virus or no match (defined as 'non-cellular'). Different cut-off values were used for classifying contigs as 'cellular' (i.e., >90, 80, 70, or 50% of the ORFs) and the results were compared to the placement of contigs into either cellular or non-cellular bins (Supplementary Fig. 1). In addition, all contigs were verified as of viral origin through VirSorter[57] using both the Virome and RefSeq databases (October 2016).

**Viral taxonomy and functional annotation**. To gain insight into the similarity of the Äspö viral contigs to known isolated viruses, all predicted proteins were compared to the NCBI viral database (ViralDB) created from all viral genomes in the NCBI RefSeq database (July 2019, downloaded using NCBI FTP: ftp://ftp.ncbi.nlm.nih.gov/) using BLASTp (v 2.7.1, e-value cut-off 0.001). In addition, network analysis of isolated, genome-sequenced bacterial and archaeal viruses, was conducted using vConTACT2 (version 0.9.2, NCBI Bacterial and Archaeal Viral RefSeq V85; reference[34]) with default settings except a Diamond e-value cut-off of 0.001. The network was visualized using Cytoscape (v3.7.2) where column 1 was selected as Source Node, column 2 as Target Node, and column 3 as Edge Attribute and duplicate edges were removed. Functional annotation was performed with hmmsearch against the PFAM database (v3.1b2, e-value cut-off 0.001, April 2018, PFAM database downloaded using PFAM FTP: ftp://ftp.ebi.ac.uk/pub/databases/Pfam/releases/Pfam), which was complemented with Diamond-BLASTp (v 0.8.26; e-value cut-off 0.001) against NCBI nr (June 2017), BLASTp (v 2.7.1, e-value cut-off 0.001) against ViralDB (January 2017), and BLASTp (v 2.7.1, e-value cut-off 0.001) against a marine viral proteome dataset[58].

**Comparison of Äspö HRL datasets and to other viral metagenomic datasets**. The relative abundance for each contig in all Äspö HRL viral metagenomes was estimated in the binning process (see above) and contigs covered with reads over >75% of the contig length were regarded as present in the given sample[36]. Reads

from the Linnaeus Microbial Observatory (LMO) viral metagenomic dataset[32] (BioProject accession number PRJNA474405) and microbial metagenomic dataset[59] (accession numbers SRS954972, SRS954970, SRS954960, SRS954956, SRS954951, SRS954969, and SRS954965) were also recruited to the Äspö HRL dataset (bowtie2 v.2.3.4.1, samtools 1.1, and bedtools 2.27.1). As the LMO datasets were sampled and extracted in the same laboratory using identical methods, chemicals, and extraction kits, it was appropriate for identifying potential contamination. As only 49 out of 4051 Äspö HRL contigs (equaling ~0.1%) occurred in any LMO viral metagenome and no Äspö HRL contig occurred in more than 13 of 25 LMO viral metagenomes, this suggests contamination during sample preparation and extraction was not an issue within the Äspö HRL dataset.

The Äspö HRL viral proteins were also clustered to the Tara Oceans viral protein clusters (/iplant/home/shared/irvirus/TOV_43_viromes/TOV_43_all_contigs_predicted_proteins.faa.gz[36]) using PC-pipe through the CyVerse interface with default settings. The Äspö HRL viral ORFs were further aligned to 16 publicly available viral metagenomic datasets through iMicrobe (downloaded using Cyberduck, version 7.0.2)[29,60–63] and three metagenomically assembled deep biosphere viral genomes[37] using BLASTp (e-value cut-off 0.001).

**Transcriptomic analysis.** Viral transcript data utilized in the present study were originally published in studies describing community RNA extracted from cells captured in either a specially designed sampling device that maintained cells under in situ conditions followed by cell capture on 0.1-μm filters[7] or directly on 0.1-μm filters placed in a high-pressure filter holder over an extended time period[6]. These two sampling regimes generated biological replicates of community RNA from the different groundwaters corresponding to MM-171.3 ($n = 3$), MM-415.2 ($n = 2$), and TM-448.4 ($n = 4$). RNA extraction and sequencing were performed in these studies using the MO BIO PowerWater RNA isolation kit and cDNA generated with the Ovation® RNA-Seq System V2 (NuGEN) and Illumina library preparation used the TruSeq Nano DNA Library Prep Kit for NeoPrep. Samples were sequenced on a HiSeq2500 with a $2 \times 126$ bp setup using 'HiSeq SBS Kit v4' chemistry. In this study, transcriptomic reads from the different boreholes were recruited against the contigs from the metagenome generated from the same borehole with bowtie2 (sampling device: v 2.2.9 and filter holder: v 2.3.3.1; both default settings). Coverage and depth were calculated with samtools (sampling device: v 0.1.19 and filter holder: v1.1) and BEDTools (sampling device: v 2.26.0 and filter holder: v 2.27.1). Metatranscriptome sequencing data are deposited in the NCBI BioProject ID PRJNA400688.

**Host affiliation.** Host prediction for reconstructed viral genomes was performed through pairwise comparison of tetramer frequency vectors[40,64] between each viral genome and a genome dataset including all bacterial/archaeal genomes available at NCBI (170378 genomes as per October 2018) and MAGs from Äspö HRL aquifers[25,65]. Tetramer frequency vectors were calculated with Jellyfish[66]. Dissimilarity of frequency vectors was calculated as mean absolute error (MAE) and reported in Supplementary Data 1. Taxonomic affiliation of putative hosts was extracted from the Genome Taxonomy database (https://gtdb.ecogenomic.org/) for NCBI genomes and assessed with the GTDB-Tk tool[40] for Äspö HRL MAGs. The network analysis described above was adapted in Cytoscape to show predicted host affiliation for the viral contigs. Comparison of contigs of viruses potentially infecting Patescibacteria, Firmicutes, Desulfobacteriota, and Archaea and with similarity to known phages infecting Gammaproteobacteria were conducted using EasyFig[67].

**Statistics and reproducibility.** The three boreholes were independently sampled and processed in duplicate giving true biological metagenome replicates[25]. Biological replicates were also independently sampled from the same boreholes as for the metagenomes using two methods for metatranscriptome analysis[6,7] giving $n = 3, 2$, and 4 for MM-171.3, MM-415.2, and TM-448.2, respectively. Statistical analysis of SSU rRNA and annotated transcripts using UniProtKB identifiers showed a high degree of similarity with the exception of one TM-448.2 outlier compared to the remaining triplicate samples[6].

**Reporting summary.** Further information on research design is available in the Nature Research Reporting Summary linked to this article.

## Data availability
Raw reads from the metagenomes[25] and metatranscriptomes[6,7] are deposited in NCBI with the BioProject IDs PRJNA279924 and PRJNA400688, respectively. Viral metagenomic datasets[32] from the Linnaeus Microbial Observatory (LMO) are available at NCBI with the accession number PRJNA474405 and LMO microbial metagenomic datasets[59] are available at NCBI with accession numbers SRS954972, SRS954970, SRS954960, SRS954956, SRS954951, SRS954969, and SRS954965. The viral contigs are available from NCBI with accession numbers MT141153 to MT145203 and are listed in Supplementary Data 1. MAGs used for host prediction and viral contigs are also available through the Zenodo repository with the identifier [data DOI: 10.5281/zenodo.3700451][68]. Complete and unedited Cytoscape figures are available in Figshare repository with the identifier [DOI: 10.6084/m6089.figshare.11590494.v11590491][35].

## Code availability
Bioinformatic pipelines used in this study are detailed in the Zenodo repository with the identifier [data DOI: 10.5281/zenodo.3700451][68]. All other bioinformatical analyses have been conducted using published pipelines that are cited within the "Methods" section.

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

## Acknowledgements

All authors gratefully acknowledge SKB for providing access to the Äspö HRL tunnel and laboratories. K.H., M.D., A.A., and S.B. thank the Swedish Research Council, Vetenskapsrådet (contracts 2013-4554, 2014-4398, 2017-04422, and 2018-04311); M.D. and K.H. are grateful to The Crafoord Foundation (contract 20130557); and M.D. thanks Nova Center for University Studies, Research and Development, and Familjen Hellmans stiftelse for funding. DNA was sequenced at the National Genomics Infrastructure hosted by Science for Life Laboratory. Bioinformatic analysis employed the Uppsala Multidisciplinary Center for Advanced Computational Science resource at Uppsala University (project b2013127).

## Author contributions

K.H. and M.D. conceived the study. K.H., M.L.-F., X.W., and M.D. prepared and processed metagenome and metatranscriptome samples. K.H., E.N., D.S., I.d.B., S.B., D.L., and A.A. performed bioinformatic analyses. K.H., S.B., and M.D. interpreted the data. K.H. and M.D. wrote the initial draft of the manuscript. All authors commented and approved the final manuscript.

## Funding

## Competing interests

The authors declare no competing interests.
