## [Peer Review File · Communications Biology]

Reviewers' comments:

Reviewer #1 (Remarks to the Author):

This manuscript characterizes viromes collected from three sites from a vast and largely unstudied biome, deep subsurface ground water. Based on metagenomics, viruses were identified at three different depths. The shallower sights were a mixture of water from the Baltic Sea and groundwater, while the deepest sight was long residing groundwater from different sources. Viral communities from the deepest samples were more similar than the shallower sample. In general, the viruses from these samples were divergent from the genomes of viral isolates and published viromic data. Most viruses from these communities were from the Caudovirales and were linked to a variety of host groups, including to the Patescibacteria and Archaea. The authors concluded that the surface communities were metabolically active while the deeper communities were essentially dormant. Several auxiliary metabolic genes were identified, including AMGs involved in Sulfur cycling, an important process in this environment.

This is interesting research, resulting in rare data on the viral communities of an extremely inaccessible environment. Although in general I found the work compelling, there remains a few issues that should be addressed. It is clear that this is an exploratory study. Nevertheless, there is enough published data on subsurface groundwater microbial communities for the authors to generate hypotheses. These should be stated. I think that the paper would be enhanced with the addition of an analysis of the Viral Assembled Genomes beyond the subset of putative viruses of the Patescibacteria (there characterization as well as their relative abundance at the three depths. I am also concerned by the fact that the transcriptomic data was not concurrently collected with the viromic data, complicating the interpretation of these data and bringing into question the conclusion that the deeper samples were dormant. Please find specific comments and questions below:

Line 53: the citation needs to be corrected

Line 62: rephrase "further"

Line 72: the citation needs to be corrected

Line 72-75: rephrase, repetitive

Line 78: minor comment on semantics- I suggest the term "virome" over "metavirome", which I find redundant

Line 77-82: I suggest going beyond the descriptive and including hypotheses. What were the researchers expectations and why?

Line 90, 91 and 93: Are the measurements of depth precise enough to show the tenths decimal place?

Line 95 Ideally the "waters of other origins" would be better characterized as this these other origins will ultimately drive viral diversity in the sample

Line 99: It should also be stated what types of viruses are removed with a 0.22 μm filtration step. Presumably viruses adsorbed to cells, intracellular and integrated viruses, and viruses larger than 220 nm in diameter.

Line 101: Why greater than 10 genes? Can this choice be justified with a reference or with supplementary data?

Line 113: Are these percentages affected by the relative depth of sequencing? Has an assessment of coverage been performed (Nonpareil, for example)

Line 146: Given the apparently high percentage of cellular contamination, it would be interesting to see a comparison with cellular metagenomes (from the Baltic Sea for example), and not uniquely viromes. I imagine the percentage of matches would be much higher.

-It would also be interesting to have a comparison that is not solely based on assembled data, for

example Libra which is based on sequence reads and does not require assembly.

Line 149: What is the estimated residence time of the seawater in the groundwater? Is this counter your expectation?

Line 168: Can the groups shared among the samples be further characterized? What constitutes the core groundwater virome based on these samples?

Line 172: Is the use of different methods a source of bias? Is the porosity to capture the cells 0.22?

Line 183: It should be pointed out that this is looking at as of yet undefined subset of the viral population, thus it is unclear if this represents the metabolic state of the entire community. In addition, this is one snapshot in time of those viruses that were actively replicating in one particular moment and may not be generally representative.

Line 290: A caveat should be added, however, that the activity of most of these viral genes has not been established

Line 311: Even after flushing, ideally there would have been some sort of blank to account for the contribution of the bore hole itself (likely impossible I assume)

Line 317: The manufacturer of the kit should be mentioned. This kit involves an amplification step that could introduce bias and should be mentioned.

Line 683: What were the criteria used to select the contigs shown in Figure 6?

Reviewer #2 (Remarks to the Author):

The manuscript "The deep terrestrial virosphere" by Holmfeldt et al. study the virome and RNA transcript from three groundwaters. Generally, the experimental design is clear. The methods used are reasonable. This manuscript updates our understanding of viruses in the terrestrial virosphere. Hence, that is an interesting paper and their findings were valuable. But several points must be clarified and addressed as presented below. Firstly, the title is improper. Deep terrestrial bio/virosphere comprises various ecosystems, such as subsurface sediment, coal mine, groundwater, mud volcano etc. Here, the authors should show the specific information about this study. In the Abstract, the interesting phenomenon "metabolic standby" was claimed. However, I did not see any analysis or discussion in the MS to support the point. A more in-depth discussion is needed.

There are also some small errors in the manuscript.

L42: Deep biosphere is not always oligotrophic.

L44: Please also cite the paper published by Bar-On et al.

L50: This sentence is confusing.

L53: Please check the ref citation throughout the MS.

L73: Podoviridae should be italic.

Response to Reviewers

Reviewer #1

This manuscript characterizes viromes collected from three sites from a vast and largely unstudied biome, deep subsurface ground water. Based on metagenomics, viruses were identified at three different depths. The shallower sights were a mixture of water from the Baltic Sea and groundwater, while the deepest sight was long residing groundwater from different sources. Viral communities from the deepest samples were more similar than the shallower sample. In general, the viruses from these samples were divergent from the genomes of viral isolates and published viromic data. Most viruses from these communities were from the Caudovirales and were linked to a variety of host groups, including to the Patescibacteria and Archaea. The authors concluded that the surface communities were metabolically active while the deeper communities were essentially dormant. Several auxiliary metabolic genes were identified, including AMGs involved in Sulfur cycling, an important process in this environment.

This is interesting research, resulting in rare data on the viral communities of an extremely inaccessible environment. Although in general I found the work compelling, there remains a few issues that should be addressed. It is clear that this is an exploratory study. Nevertheless, there is enough published data on subsurface groundwater microbial communities for the authors to generate hypotheses. These should be stated.

Reply: Thank you for your suggestion. We have now been more explicit about the hypotheses and present these in the final paragraph of the introduction accordingly: “Our hypotheses were that: (1) due to similar geochemistry and age²⁵, the two Äspö HRL modern marine groundwater viromes are more similar to each other than to the thoroughly mixed water and (2) that similarly to the bacterial community^{6,7}, the more shallow virome is more active than the deeper groundwaters.” (lines 83 to 86). These conclusions have also been updated to reflect these hypotheses: “Despite that the two modern marine groundwaters have similar water characteristics, the MM415.2 and TM-448.2 viromes were more similar to each other than MM-171.3 to MM415.2 and thus, refuting hypothesis one. The reason for the higher resemblance of viromes from MM415.2 and TM-448.2 is yet unclear but may reflect the greater depth from the surface for these two groundwaters. The presence of RNA transcripts predominantly in the MM-171.3 validated hypothesis two of an actively replicating microbial community in the shallow groundwater compared to the community in the thoroughly mixed ground water, which is in ‘metabolic standby’.” (lines 332-340).

I think that the paper would be enhanced with the addition of an analysis of the Viral Assembled Genomes beyond the subset of putative viruses of the Patescibacteria (there characterization as well as their relative abundance at the three depths).

Reply: Complete genome analyses of all Viral Assembled Genomes are presented in the Supplemental Data files (previously supplemental tables), e.g. Supplemental Data 1 contain main information of all contigs, including length, ORF numbers, overview of matches, as well as relative abundance in all metagenomes and activity (seen as transcript read recruitment). In Supplemental Data 4, the functional annotation of each individual ORF is given. We now point out this information more clearly with references to the Supplemental Data files, e.g. line 175, 205, 225, 284, and 293). In addition, we have produced figures of selected Viral Assembled Genomes for Firmicutes, Gammaproteobacteria, Desulfobacteriota, and Archaea. These results have been included in the text (e.g. lines 227, 252-255, 272-275) along with new Supplemental Figures S7 to S10.

I am also concerned by the fact that the transcriptomic data was not concurrently collected with the viromic data, complicating the interpretation of these data and bringing into question the conclusion that the deeper samples were dormant.

Reply: We agree with the reviewer that it would have been optimal to carry out viral metatranscriptomes at the same time as the metagenomes. However, published data (see

<https://doi.org/10.1128/mBio.01470-19>) demonstrate that bacterial metatranscriptomes sampled several years apart are statistically similar to each other for the same modern marine and thoroughly mixed groundwaters as presented in this study. We argue that this statistical analysis supports a stable Fennoscandian Shield deep biosphere community over the time range of a few years and that the metatranscriptomes in this study can be validly compared to the original viral metagenomes.

1. Line 53: the citation needs to be corrected

Reply: Thank you for your comment. The text has now been changed to “*as previously reported*¹²” (line 53).

2. Line 62: rephrase “further”

Reply: “further” has been changed to “additional” (line 62).

3. Line 72: the citation needs to be corrected

Reply: We have now changed the sentence so that the reference is in the beginning: “*Kyle et al.*²⁶ reported that...” (line 73).

4. Line 72-75: rephrase, repetitive

Reply: The sentence has been simplified and shortened to remove the repetition. It now reads “*and include a lytic Podoviridae virus infecting the sulfate reducing bacterium Pseudodesulfovibrio aespoensis (formerly Desulfovibrio aespoensis)*²⁷.” (lines 75-76).

5. Line 78: minor comment on semantics- I suggest the term “virome” over “metavirome”, which I find redundant

Reply: “Metaviromes” has been changed to “viromes” in two places (lines 30 & 80).

6. Line 77-82: I suggest going beyond the descriptive and including hypotheses. What were the researchers expectations and why?

Reply: We have generated hypotheses and they are presented in the final paragraph of the introduction accordingly: “*Our hypotheses were that: (1) due to similar geochemistry and age*²⁵, the two Äspö HRL modern marine groundwater viromes are more similar to each other than to the thoroughly mixed water and (2) that similarly to the bacterial community^{6,7}, the more shallow virome is more active than the deeper groundwaters.” (lines 83-86). The conclusions have also been updated to reflect these hypotheses: *Despite that the two modern marine groundwaters have similar water characteristics, the MM415.2 and TM-448.2 viromes were more similar to each other than MM-171.3 to MM415.2 and thus, refuting hypothesis one. The reason for the higher resemblance of viromes from MM415.2 and TM-448.2 is yet unclear but may reflect the greater depth from the surface for these two groundwaters. The presence of RNA transcripts predominantly in the MM-171.3 validated hypothesis two of an actively replicating microbial community in the shallow groundwater compared to the community in the thoroughly mixed ground water, which is in ‘metabolic standby’.*” (lines 332-340).

7. Line 90, 91 and 93: Are the measurements of depth precise enough to show the tenths decimal place?

Reply: Yes, the Äspö HRL is used to study the potential geological deposition of nuclear waste and is one of the most carefully characterized deep biosphere sites in the world.

8. Line 95 Ideally the “waters of other origins” would be better characterized as this these other origins will ultimately drive viral diversity in the sample.

Reply: More details of the infiltrating waters have been added as requested: “*such as glacial meltwater from the most recent Pleistocene continental ice sheet along with water infiltrating from the Littorina Sea a few thousands of years ago*” (lines 101-103).

9. Line 99: It should also be stated what types of viruses are removed with a 0.22 μm filtration step. Presumably, viruses adsorbed to cells, intracellular and integrated viruses, and viruses larger than 220 nm in diameter.

Reply: This caveat has been added to the text: "In addition, this filtration step will have removed viruses adsorbed to cells, intracellular and integrated viruses, and viruses larger than 0.22 μm ." (lines 110-111).

10. Line 101: Why greater than 10 genes? Can this choice be justified with a reference or with supplementary data?

Reply: The choice of ten genes was made to be able to have a large enough number of genes to be able to include some matches to NCBI but still small enough to include small viruses (phiX174 has for example an average of 11 genes). We are aware that some viruses might have a smaller number of genes and would therefore be missed. However, the results we gained are still very similar to the results from VirSorter, where a recommended genome length is 10 kb, even though they also see quite good results down to 3 kb.

11. Line 113: Are these percentages affected by the relative depth of sequencing? Has an assessment of coverage been performed (Nonpareil, for example).

Reply: The percentages are number of reads (in bp) mapping to a certain organism group divided with the dataset size, thus accounting for the depth of sequencing. A Nonpareil assessment has now been performed and presented in the results accordingly: "These percentages were unaffected by the sequencing coverage assessed using Nonpareil as the metagenomes showed metagenomic coverage of 66-86% (Supplementary Table S1)." (lines 126-128). The methods are provided accordingly: "Nonpareil was run on all metagenomes individually (version 3.304, settings: -T kmer -f fastq) to estimate the sequencing coverage of the community⁵³" (lines 369-370).

12. Line 146: Given the apparently high percentage of cellular contamination, it would be interesting to see a comparison with cellular metagenomes (from the Baltic Sea for example), and not uniquely viromes. I imagine the percentage of matches would be much higher. It would also be interesting to have a comparison that is not solely based on assembled data, for example Libra which is based on sequence reads and does not require assembly.

Reply: Thank you for your comment. However, there is only a small overlap of cellular metagenomic reads to the assembled contigs in the Äspö HRL dataset. We have now recruited reads from seven metagenomes from the Linnaeus Microbial Observatory in the Baltic Sea (all from 2012 and sampled at the same date or in the close vicinity to when the viral metagenomes from LMO were sampled). On average, only 3.6% of contigs recruited any reads and < 0.2% of contigs recruited any reads with >75% contig coverage. This information has now been added in the results and discussion section accordingly "Despite a high proportion of cellular organisms within the Äspö HRL viral metagenome, recruitment of reads from seven Baltic Sea microbial metagenomes to all Äspö HRL contigs showed that <0.2% of contigs were covered with reads covering >75% of the contig length (Supplementary Data S3, sheet recruit. LMO cell. metag.)." (lines 159-163). The methods are described on line 405-406 and the results is presented in Supplementary Dataset S3.

We have chosen to not do comparison of not assembled data, as in this manuscript we wish to be able to separate viral data from microbial data, which would be difficult if using non-assembled data given the cellular contamination from ultrasmall cells.

13. Line 149: What is the estimated residence time of the seawater in the groundwater? Is this counter your expectation?

Reply: The residence time of <20 years has been added to the manuscript accordingly: "This finding was remarkable since MM-171.3 and MM-415.2 groundwaters have a residence time of <20 years and are fed by infiltration of Baltic Sea water likely to transport also microorganisms into this deep

groundwater²⁴ (lines 165-168). The lack of viral sequences that are also identified in the Baltic Sea is further supported by the unpublished amplicon study mentioned in the previous reply and is surprising as the marine deep biosphere is known to consist of selected populations from the surface microbes. This has also been added: *“This suggests a cell sorting process has occurred between marine surface water microorganisms, through shallow sediments that consist of a selection of microbes found in surface sediments³⁷, to deep biosphere communities.”* (see lines 168-170).

14. Line 168: Can the groups shared among the samples be further characterized? What constitutes the core groundwater virome based on these samples?

Reply: The core virome shared between the three water types has been analyzed with text added in the results and discussion accordingly: “The core viral community (contigs occurring in all three groundwaters) were highly diverse and were included in network-clusters ranging from only core-clusters to clusters composed of viruses both in the core and those occurring in only one groundwater (Fig. 1 and Supplementary Figs. S4 & S5). Several of the most abundant viruses in the Äspö HRL were present within the core, but often they were only highly abundant in the groundwater they were originally assembled from and occurred in low abundances in the other groundwaters. For instance, the core MM171B00549, MM171A01743, and MM171B01602 viruses were highly abundant in MM-171.3 (Fig. 1, cluster 1) and TM448A02179 was highly abundant in TM-448.2 (Fig 1, cluster 2).” (see lines 184-192). In addition, a new figure 1 (accompanied with a supplemental figure S2) highlighting the groundwaters each contig is present in (thus visualizing the core clearly) has replaced the previous figure 1, which showed which groundwater the contig originated from (was assembled from). The old figure 1 has been moved to the supplemental files, Fig S4).

This is the new figure 1:

Fig. 1 | Network displaying Äspö HRL contigs and viral isolates highlighting connections between groundwaters. A selected portion of the network analysis of the contigs from the three groundwaters along with sequences from bacterial and archaeal viral isolates in the NCBI RefSeq database (black nodes). Color coding of nodes represents the presence of the contigs in the groundwaters based on metagenome read recruitment. Boxes marked with letters refer to phages that mainly infect Gamma-, Beta-, and Alphaproteobacteria (A1 & A2); Firmicutes (B); and Bacteroidetes (C1 & C2). Boxes marked with numbers refer to clusters consisting of viral contigs both in the core and in individual groundwaters. Node sizes for Äspö HRL viral contigs are based on bp reads recruited divided by the contig size in kb and the metagenome size in Mb (smallest circle <50, medium circle 50 – 100, and largest circle >100 bp mapped/kb of contig/Mb of metagenome) for the metagenome from which the

contig originated (see Supplemental Fig. S4 for information regarding the groundwater type where a specific contig originated). Reference viral isolates in the NCBI RefSeq database are all presented using the smallest circle size and do not represent abundance. The complete and unedited figure is available in Supplementary Fig. S2.

Legend for Supplemental figure S2:

Supplementary Fig. S2. Network displaying Äspö HRL contigs and viral isolates highlighting connections between groundwaters. Cytoscape file for network analyses with presence of viruses within the three different groundwaters. Color code for the groundwater in which the contigs have been detected in are given under Fill color. Note: Some unconnected networks have been moved due to spatial considerations in Fig. 1 and contig names in the Cytoscape file are defined as MM-171.3 = MM, MM-415.2 = UM, and TM-448.2 = OS.

15. Line 172: Is the use of different methods a source of bias? Is the porosity to capture the cells 0.22?

Reply: That the methods used do not result in a bias has been described in Lopez-Fernandez et al. (see: mBio 10:e01470-19. <https://doi.org/10.1128/mBio.01470-19>). Both methods used a membrane porosity of 0.1 μm for cell capture. This has now been clarified accordingly: “Viral transcript data utilized in the present study were originally published in studies describing community RNA extracted from cells captured in either a specially designed sampling device that maintained cells under in situ conditions followed by cell capture on 0.1 μm filters⁷ or directly on 0.1 μm filters placed in a high-pressure filter holder over an extended time period⁶.” (lines 421-425.)

16. Line 183: It should be pointed out that this is looking at as of yet undefined subset of the viral population, thus it is unclear if this represents the metabolic state of the entire community. In addition, this is one snap shot in time of those viruses that were actively replicating in one particular moment and may not be generally representative.

Reply: We have added text to the results and discussion explaining how this was a ‘snapshot’ of a subset of the community and that we cannot rule out the presence of unidentified viruses active at the time of cell capture. The text currently read: “These metatranscriptomes represent ‘snapshots’ sampled at different time points than when the viral metagenomes were generated and they are only compared to a subset of the entire viral community (represented by the metagenomically assembled contigs) lacking e.g., prophages removed in the metagenomes by cell capture on the 0.2 μm filters. Therefore, we cannot rule out that additional, unidentified viruses in the MM-415.2 and TM-448.2 groundwaters were active at the time of cell capture.” (lines 205-210).

17. Line 290: A caveat should be added, however, that the activity of most of these viral genes has not been established.

Reply: We have added qualifiers in two places in the indicated paragraph that only a part of the viral community is active (see lines 322 and 326).

Line 311: Even after flushing, ideally there would have been some sort of blank to account for the contribution of the borehole itself (likely impossible I assume).

Reply: While we agree with the reviewer that this would be desirable, the reviewer is also correct that it is impossible to take a sample accounting for a borehole extending into the bedrock.

Line 317: The manufacturer of the kit should be mentioned. This kit involves an amplification step that could introduce bias and should be mentioned.

Reply: The manufacturer has been added (line 358) and the amplification caveat is added so it now reads “However, it should be noted that the ThruPlex DNA-seq Kit includes an amplification step that

can potentially add bias to the sequencing libraries.” (lines 361-363).

Line 683: What were the criteria used to select the contigs shown in Figure 6?

Reply: Thank you for your comment. Due to space limitations we have chosen to display the Patescibacteria virus contigs for which a taxonomic rank below Patescibacteria could be confirmed as well as any contigs with similarity to those. This information has now been added to the figure legend accordingly: “Selected viral contigs suggested by kmer analysis to infect Patescibacteria where lower taxonomic ranks could be determined as well as contigs similar to those in the three Äspö HRL groundwaters.” (lines 798-800).

Reviewer #2

The manuscript “The deep terrestrial virosphere” by Holmfeldt et al. study the virome and RNA transcript from three groundwaters. Generally, the experimental design is clear. The methods used are reasonable. This manuscript updates our understanding of viruses in the terrestrial virosphere. Hence, that is an interesting paper and their findings were valuable. But several points must be clarified and addressed as presented below.

Firstly, the title is improper. Deep terrestrial bio/virosphere comprises various ecosystems, such as subsurface sediment, coal mine, groundwater, mud volcano etc. Here, the authors should show the specific information about this study.

Reply: Thank you for your comment. We have altered the title to more accurately reflect the study is of the Fennoscandian Shield deep biosphere. The new title is “The Fennoscandian Shield deep terrestrial virosphere” (line 3).

In the Abstract, the interesting phenomenon “metabolic standby” was claimed. However, I did not see any analysis or discussion in the MS to support the point. A more in-depth discussion is needed.

Reply: The term “metabolic standby” has now been explained in the introduction: “Furthermore, RNA transcripts from Äspö HRL groundwaters with higher relative concentrations of organic carbon showed an active community with a range of metabolic strategies. In contrast, the low organic carbon water has the potential to carry out translation but only replicates when a carbon and energy source is intermittently available, a state defined as ‘metabolic standby’^{6,7}” (lines 69-73). We also discuss how the viral community in the MM-171.3 is actively replicating while viruses in the MM-415.2 and TM-448.2 groundwaters largely lack the required replicating hosts in order to be active: “However, as the MM-171.3 groundwater has an active, replicating microbial community while the TM-448.2 community appears to be in “metabolic standby”⁷, the most plausible explanation is that the viral community in the MM-171.3 is also actively replicating while viruses in the MM-415.2 and TM-448.2 groundwaters largely lack the required replicating hosts in order to be active” (lines 211-215).

1. L42: Deep biosphere is not always oligotrophic.

Reply: We agree with the reviewer and have altered the text to “Despite being predominantly highly...” (line 43).

2. L44: Please also cite the paper published by Bar-On et al.

Reply: The reference has been added (line 44).

3. L50: This sentence is confusing.

Reply: The sentence has been edited for clarity accordingly: “Deep terrestrial biosphere viral abundances range from 104 to 106 particles mL⁻¹ with a maximum virus:bacteria ratio of 7:1, as previously reported¹²” (lines 51-53).

4. L53: Please check the ref citation throughout the MS.

Reply: This has now been checked and correct e.g. at line 53 and line 73. In addition, the reference list has been checked thoroughly.

5. L73: Podoviridae should be italic.

Reply: This has been changed according to the reviewer's suggestion (line 75).

REVIEWERS' COMMENTS:

Reviewer #1 (Remarks to the Author):

The authors have adequately addressed the points I have raised and I have no further questions or comments.

Reviewer #2 (Remarks to the Author):

No further comments.